# Conceptual Framework for the Psychosocial Support of Nurses Caring for Patients Diagnosed with COVID-19 Infection in North West Province, South Africa

**DOI:** 10.3390/ijerph20065078

**Published:** 2023-03-14

**Authors:** Joan Mologadi Dikobe, Miriam Mmamphamo Moagi, Leepile Alfred Sehularo

**Affiliations:** School of Nursing, Faculty of Health Sciences, North West University, Mahikeng 2735, South Africa

**Keywords:** psychosocial support, nurses, caring, COVID-19, patients, conceptual framework

## Abstract

Introduction: A conceptual framework provides the detailed components or concepts identifying the interrelationships in and across a project’s components. Due to a lack of psychosocial support, nurses caring for patients diagnosed with COVID-19 are physically, psychologically and socially affected. However, there are no conceptual frameworks for the psychosocial support of nurses caring for patients diagnosed with COVID-19 infection in North West Province, South Africa. The purpose of this study was to develop a conceptual framework for the psychosocial support of these nurses. Design: A qualitative, descriptive phenomenological and contextual research design was followed to conduct this study. Six questions were used to classify concepts and develop the proposed framework. These six crucial questions are based on the agent, recipient, context, procedure, dynamics and terminus. Findings: The results of the framework involved the mobilisation of effective managerial support, the provision of adequate human medical healthcare resources and the mobilisation of support from nurses working in non-COVID wards and family members in the provision of psychological support systems (procedure). The newly developed conceptual framework aims to support nurses caring for patients diagnosed with COVID-19 infection in North West Province (terminus) and to improve their wellbeing. Conclusion: The developed framework provides information that can assist nurses in providing quality care to patients. Contribution: The framework will provide solutions for healthcare institutions to respond effectively to similar pandemics in the future, improving the psychosocial wellbeing of nurses caring for patients diagnosed with COVID-19.

## 1. Introduction

Nurses play an essential role in providing care, treatment and rehabilitation to patients diagnosed with COVID-19 infection. Their psychosocial wellbeing and safety are crucial for ensuring safe patient care, as well as for the control of infectious diseases such as COVID-19 (Dikobe, Moagi and Sehularo, 2023) [1]. In a study conducted by Bakar and Ramli (2020) [2], the psychosocial wellbeing of nurses as well as their morale and work performance were shown to be jeopardised by working long hours under uncomfortable and stressful conditions. There is emerging evidence that nurses caring for patients diagnosed with COVID-19 infection have greater risks of being affected psychologically, and as a result, they develop mental health conditions such as anxiety, depression, insomnia and stress (Aughterson et al., 2021 [3]). Supporting the mental health of nurses is a critical part of the public health response to the pandemic, because nurses experience psychological and social pressure when caring for patients diagnosed with COVID-19 infection (Walton et al., 2020) [4]. In this study, the framework developed by Dickoff et al. (1968) [5] is used to create a conceptual framework for the psychosocial support of nurses caring for patients diagnosed with COVID-19 infection.

A conceptual framework represents a network of interlinked concepts and can provide a structure for understanding a phenomenon or subject (Glover et al., 2020) [6]. According to Bordage (2009) [7], a conceptual framework represents the ways in which complex things work. The same author indicates that it is used to guide a work based on best practices. A conceptual framework provides the detailed components or concepts that identify the interrelationships in and across a project’s components (Springer, 2020) [8]. The aim of the conceptual framework is to classify, define and organise ideas and to determine the relationships between concepts (Turan et al., 2022) [9]. The aim of the newly developed framework in this study is to support those nurses caring for patients diagnosed with COVID-19 infection in North West Province (NWP), South Africa.

There are several definitions of a conceptual framework. Jabareen (2009) [10] defines a conceptual framework as a network or “a plane” of interlinked concepts that together provide a comprehensive understanding of a phenomenon or phenomena. The author further indicates that the concepts that constitute a conceptual framework support one another, articulate the respective phenomena and establish a framework-specific philosophy. According to Springer (2020) [8], these concepts are logically developed and organised to support an overall framework. Meanwhile, Rogers (2016) [11] states that a conceptual framework involves reading critically and making connections or integrating and synthesising existing works related to concepts to understand the theoretical and practical contexts behind them. Polit and Beck (2021) [12] assert that frameworks are often implicit, without being formally described. The same authors further state that without an articulated conceptual framework, it is difficult to determine the researchers’ thoughts. The conceptual framework allows researchers to make reasoned and defensible choices about the topics or themes to be explored in search of new contexts (Ravitch and Riggan, 2016) [13]. A conceptual framework should be used to represent the comprehensive linkages between concepts: in this case, the themes and categories that emerge with regard to the psychosocial support needed for nurses caring for patients diagnosed with COVID-19 infection in the NWP.

Globally, the psychosocial wellbeing and support of nurses caring for patients diagnosed with COVID-19 has been a concern (Billings et al., 2020) [14]. Xu et al. (2021) [15] added that protecting nurses’ mental wellbeing by providing adequate psychosocial support during the COVID-19 pandemic has been identified as crucial. 

The literature reviewed for this study revealed that there are no existing conceptual frameworks for the psychosocial support of nurses caring for patients diagnosed with COVID-19 infection in the NWP of South Africa. Thus, there is a need for the development of a such a conceptual framework. 

### Purpose of the Study

The purpose of the study was to develop a conceptual framework for the psychosocial support of nurses caring for patients diagnosed with COVID-19 infection in the North West Province of South Africa. 

## 2. Materials and Methods

### 2.1. Study Design

A qualitative, descriptive phenomenological and contextual research design as explained by Polit and Beck (2021) [12] was followed in developing a conceptual framework. This study was conducted in three distinct phases, namely: an integrative literature review, an empirical phase and the development of the conceptual framework. The findings of the integrative literature review and empirical phases inform the development of the framework.

Phase 1: Integrative literature review (ILR)

An integrative literature review was used to explore published evidence regarding the psychosocial support guidelines for nurses caring for patients diagnosed with COVID-19 infection. Databases that were consulted included Google Scholar, Science Direct, EBSCOhost, Medline, Emerald and Scopus. Articles published between 2019 and 2022 were explored to improve the credibility of the review (Dikobe et al., 2023 [1], see Table 1). 

Phase 2: The empirical phase

The empirical phase of this paper is comprised of two steps using qualitative research methods. Step 1 explores and describes the lived experiences of nurses caring for patients diagnosed with COVID-19 infection. Step 2 explores and describes the psychosocial support needed for nurses caring for patients diagnosed with COVID-19 infection.

### 2.2. Population and Sampling

The study population was comprised of nurses caring for patients diagnosed with COVID-19 infection in the NWP of South Africa, including registered professional nurses, enrolled nurses and enrolled nursing auxiliary. The target population included nurses who are registered with the South African Nursing Council (SANC) and are currently employed in COVID-19 sites in NWP with more than one month of experience in caring for patients diagnosed with COVID-19 infection in NWP. A non-probability sampling technique was used to obtain a sample for the study.

### 2.3. Data Collection

Data were collected through semi-structured virtual focus group discussions (FGDs) via Google Meet in which an interview guide was used. A voice recorder and field notes were used as data collection techniques. Broad questions were asked of all participants, including: “What are your experiences in caring for patients diagnosed with COVID-19 infection?” and “What type of support do you need when you care for patients diagnosed with COVID-19 infection?” This was followed by probing questions based on the responses. Data were collected until data saturation was reached at the fourth focus group discussion.

### 2.4. Setting

The study was conducted at four selected public hospitals in which patients diagnosed with COVID-19 infection are cared for in the North West Province of South Africa.

### 2.5. Recruitment of Nurses

After receiving the approval letters to conduct the study, the researcher sent recruitment materials to the nursing service manager via email to share with nurses through their notice boards and through WhatsApp. Participants who were interested in taking part in the study informed the nursing manager. The nursing manager emailed the names and contact numbers of those nurses who were interested in participating in the study to the researcher. Appointments were made with the nurses to be interviewed virtually using FGD through Google Meet at a time that was convenient for them. The researcher made use of a virtual link when obtaining informed consent from the participants.

### 2.6. Ethical Approval

This manuscript was part of a doctoral study that was approved by the North-West University Health Research Ethics Committee (NWU-HREC) (Reference Number: NWU-00309-21-A1). The study was also approved by the North West Provincial Department of Health (NWP DoH), as well as the chief executive officers (CEOs) from which the data were collected. Permission to collect data was also obtained from the nurses. 

### 2.7. Trustworthiness

Trustworthiness was ensured through credibility, dependability, confirmability and authenticity, as described by Polit and Beck (2021) [12]. Credibility was ensured by consulting relevant databases related to the psychosocial support of nurses caring for COVID-19 patients. Credibility was further maintained by prolonged engagement with nurses during the empirical phase. Dependability was ensured by using an audio recorder for verification. Confirmability was achieved by using an audit trail as well as by writing down field notes during virtual focus group discussions. 

## 3. Results

Data saturation was achieved with a sample of four focus group discussions. A total of 26 nurses participated in the study. Participants were nurses who signed consent forms (Dikobe et al. (2023) [24] and see Table 2).

The following two themes and six categories emerged from the ILR phase.

Theme 1: Psychological support

The COVID-19 pandemic has had significant negative impacts on nurses’ psychological health, fostering issues such as anxiety, depression, sleep disturbances and burnout. This indicates the necessity of providing psychological support for nurses caring for patients diagnosed with COVID-19 infection (Alnazly et al., 2021) [16]. The following sub-themes emerged:Employee Assistance Program (EAP)

The findings of this ILR revealed that professional psychological counselling could potentially alleviate psychological illnesses, provide a theoretical basis for promoting EAP psychological services and demonstrate ways in which to reduce pressure on nurses caring for patients diagnosed with COVID-19 (Xu et al., 2021) [15].

Pastoral counselling

According to the study conducted by Chew et al. (2020) [18], many nurses found prayer and support from the religious community to be helpful during the COVID-19 pandemic.

Psychotherapy

The results of the literature search showed that supportive psychotherapy is good for every setting, including emergency settings, and for enhancing coping mechanisms (Shatri et al., 2021) [23].

Theme 2: Social support

Social support is necessary as a coping mechanism to decrease nurses’ psychological distress and promote positive feelings (Alnazly et al., 2021) [16]. The following sub-themes emerged:Family support

Muller et al. (2020) [20] reported that nurses utilised support from family during the COVID-19 pandemic as a major emotional and motivational factor to continue caring for COVID-19 patients.

Peer and colleagues’ support

A study conducted by Rathnayake et al. (2021) [21] found that working together with colleagues in a harmonious working environment can help to relieve psychological stress, especially during the time of COVID-19.

Management support

A study conducted by Sehularo et al. (2021) [22] found the importance of hospital administration’s provision of psychological support as a crucial factor in the nurses’ ability to overcome the challenges caused by COVID-19.

The following two themes and four categories emerged from step 1 of the empirical phase. 

Theme 1: Nurses’ knowledge of COVID-19 (Dikobe et al., 2022 [24]). The following categories detail the experiences of nurses caring for COVID-19 patients:

Nurses’ conceptualisation of COVID-19

The study has identified that nurses seem to have an understanding of what COVID-19 is and also the symptoms of COVID-19.


*“COVID-19 is an infectious disease caused by virus known as SARS-CoV 2 and MERS. It affects the respiratory systems (P5)”.*



*“COVID-19 is an infectious communicable disease that can be transmitted from one person to another (P1)”.*


Theme 2: Nurses’ experiences of caring for patients diagnosed with COVID-19 infection (Dikobe et al., 2022 [24]).

The findings of the study revealed that nurses have different experiences in caring for COVID-19 patients.

Nurses have distinct physical experiences in caring for patients diagnosed with COVID-19 infection.-Nurses verbalised that as a result of caring for patients diagnosed with COVID-19, they ended up developing physical problems, and some of the nurses indicated that they ended up having COVID-19 infection.


*“It wasn’t easy seeing people dying day in and day out in your hands, we would be resuscitating patients with COVID-19 and most of us ended up having COVID-19. I was one of them (P3)”.*


Nurses have distinct psychological experiences in caring for patients diagnosed with COVID-19 infection.-Nurses reported that they are afraid of being infected with COVID-19, and they also felt that they would die of COVID-19.


*“it was very traumatising and we were not sure whether we were going to get the infection or not. We were always on edge when we care for patients with COVID infection (P7)”.*



*“but it was not easy because at the same time we were scared that what if I get it or what if I don’t see that I have it and infect somebody that is closer to me and seeing the patients suffer in that way we were scared (P4)”.*


Nurses have distinct social experiences in caring for patients diagnosed with COVID-19 infection.-The findings of the study further revealed that nurses were stigmatised and socially isolated, because they were seen as carriers of COVID-19 infection.


*“the hospital staff were afraid of us, they feared us, and they didn’t want us near them when we were meeting them in the corridors of the hospital (P2)”.*


The following two themes and four categories emerged from step 2 of the empirical phase. 

Theme 1: Nurses’ challenges when caring for patients diagnosed with COVID-19 infection (Dikobe et al., 2022) [25]. The following categories detail the challenges of nurses caring for COVID-19 patients.

Nurses experienced challenges in caring for patients diagnosed with COVID-19 infection.-The findings of this study revealed that because COVID-19 is a new disease, there is no specific treatment for COVID-19, and it is difficult to deal with its complications and complexities.


*“People used concoctions, we had lots of renal failures. Patients were drinking traditional concoctions because there were no cure (P5)”.*


Nurses experienced challenges in obtaining support when caring for patients diagnosed with COVID-19 infection.-Participants indicated that it was difficult for them to obtain psychological support, and they further indicated that the management team was unproductive, as they never checked on the staff and/or referred them to occupational health and safety (OHS) for investigations such as X-rays.


*“we do not go for the x-ray, we have to take ourselves to the private doctor to check x-ray. You cough and cough and they give you only 2 days off (P2)”.*


Theme 2: Psychosocial support needed by nurses caring for patients diagnosed with COVID-19 (Dikobe et al., 2022) [25]. The following categories detail the support needed by nurses caring for COVID-19 patients.

Psychological and physical health support is needed by nurses who are caring for patients diagnosed with COVID-19.


*“We needed to be attended to by our managers. We need them to hear our cry if we do not have the equipment they should supply…if we are sick they must hear us and take care of us (P2)”.*


Resources are needed by nurses who are caring for patients diagnosed with COVID-19.


*“We also need to be supported with PPEs, there is a serious shortage (P3)”.*


## 4. Discussion

The results of this study are presented and discussed according to the three phases presented above, namely: the integrative literature review, the empirical phase and the development phase.

Using the ILR method, the researchers explored and described the published articles regarding psychosocial support for nurses caring for patients diagnosed with COVID-19 infection.

Supporting nurses psychologically is essential to preserving their health in the short and long term, particularly when occupational stress levels are so high (Maben and Bridges, 2020) [26]. The first step of the empirical phase explored and described the lived experiences of nurses caring for patients diagnosed with COVID-19 infection. Secondly, the kinds of psychosocial support needed by nurses caring for patients diagnosed with COVID-19 infection have been explored and described. 

Nurses reported that they have developed comorbidities such as hypertension since they started caring for COVID-19 patients. These findings are supported by Fang et al. (2021) [19], who reported hypertension to be a prevalent comorbidity among nurses caring for COVID-19 patients. Semo and Frissa (2020) [27] added that nurses complained of having headaches and muscle pain, so they started using analgesics. A study conducted by Walton et al. (2020) [4] revealed that the lack of specific treatment for COVID-19 and increased positive cases of and fatalities due to COVID-19 may have increased the mental health burden of nurses. 

### 4.1. Development of a Conceptual Framework

The results of the ILR and empirical phases were merged to develop a conceptual framework for the psychosocial support of nurses caring for patients diagnosed with COVID-19 infection in NWP, South Africa. 

According to Jabareen (2009) [10], grounded theory is adequate for conceptual framework building, because it involves the use of coding paradigms to ensure conceptual development. The same author further states that this is a research method aimed at the discovery of theory from systematically obtained data. Grounded theory is also contextual, procedural and inductive (Jabareen, 2009) [10]. In this study, the concepts were deduced from the themes and categories of the integrative literature review and empirical phase of the paper by asking and answering the following six questions proposed by Dickoff et al. (1968) [5] according to Practical Orientation Theory. 

The crucial questions were based on the following:Who is the agent of the framework (agent)?Who is the recipient of the framework (recipient)?In which context will the framework be implemented (context)?How will the framework be implemented (procedure)?What are the dynamics of the framework (dynamics)?What will be the endpoint of the framework (terminus)?

### 4.2. Description of the Structural Presentation of the Conceptual Framework

The description of the conceptual framework provides a process for the development of the conceptual framework for the psychosocial support of nurses caring for patients diagnosed with COVID-19 infection in NWP. Figure 1 provides a description of the conceptual framework for the psychosocial support of nurses caring for patients diagnosed with COVID-19 infection in NWP according to Dickoff et al. (1968) [5]. The process of developing the above conceptual framework was influenced by the six crucial survey list questions. The process started with the “Agent” and ended with the “Terminus” (Dickoff et al., 1968) [5]. The concepts (agent, recipient, context, procedure, dynamics and terminus) are discussed in the study.

### 4.3. Relevance and Objective of the Conceptual Framework

The conceptual framework can facilitate psychosocial support for the nurses caring for patients diagnosed with COVID-19 infection in NWP. Therefore, the conceptual framework can provide clear information on the ways in which psychosocial support for the nurses caring for patients diagnosed with COVID-19 infection can be carried out. The information in the conceptual framework can also benefit the hospitals caring for COVID-19 patients as well as policymakers. In addition, the adoption of this conceptual framework can improve clinical practice and the quality of care for patients diagnosed with COVID-19.

### 4.4. Assumptions of the Conceptual Framework

The themes and categories that emerged from the integrative literature review and empirical phases of this study were merged to develop this conceptual framework. Nurses must receive psychosocial support when caring for patients diagnosed with COVID-19 infection. Considering the fact that nurses are experiencing fear, anxiety, depression and stigmatisation, this necessitates the development of a conceptual framework for psychosocial support for the nurses caring for patients diagnosed with COVID-19 infection.

Responses to the six crucial questions proposed by Dickoff et al. (1968) [5] on Practical Orientation Theory are explained in the following paragraphs.

#### 4.4.1. Agent: Who Is the Agent of the Framework?

The first concept on the survey list of Dickoff et al. (1968) [5] is the agent. An agent of the framework refers to a person who moves the practice towards a goal by ensuring effective psychosocial support for nurses caring for patients diagnosed with COVID-19 infection. Thus, these agents have a direct impact on the psychosocial support for nurses caring for patients diagnosed with COVID-19 infection. Agents have responsibilities in enabling psychosocial support for nurses caring for COVID-19 patients. The list of agents includes the following: hospital managers, psychologists, social workers, nurses working in non-COVID wards and family members. These agents must understand their roles in supporting the nurses caring for COVID-19 patients. 

Hospital managers, as one of the agents, should provide a comprehensive support system for nurses, as nurses experience fear when caring for COVID-19 patients. Managers should ensure that they are present in the wards and provide psychological support by recognising staff, listening to their experiences and creating a safe environment, as this allows nurses to raise concerns. Management should provide nurses with material resources such as personal protective equipment (PPE), treatment and ventilators. There should also be enough staff (human resources) when caring for COVID-19 patients.

Nurses should be provided with in-service training regarding COVID-19, because it is a new disease. They should also be trained on the usage of equipment or machines used to treat COVID-19 patients. Prophylactic treatment or immune boosters should be provided to nurses caring for COVID-19 patients as appropriate according to evidence-based practice to boost their immune systems, because nurses are at risk of being infected. Furthermore, agents such as family members and nurses working in non-COVID wards should understand the emotional challenges that nurses caring for COVID-19 patients are facing and support and encourage them where necessary. 

Psychologists should be readily available in hospitals to help nurses who are caring for patients diagnosed with COVID-19. They should also assist nurses in learning to cope with stressful situations and perform psychotherapy, including individual therapy and group therapy. 

Social workers should be available in hospitals to support nurses caring for COVID-19 patients. Several social factors, such as decreased social support, seeing relatives/friends diagnosed with COVID-19 and colleagues’ readiness to cope with the COVID-19 outbreak, have increased the perceived threat of COVID-19, and the long working hours in COVID-19 wards have increased nurses’ burnout (Galanis et al., 2021) [28]. Therefore, social support is critical for nurses in the fight against epidemics.

#### 4.4.2. Recipient: Who Is the Recipient of the Framework?

The second concept on the survey list of Dickoff et al. (1968) [5] is the recipient. Recipients of this conceptual framework are the nurses caring for patients diagnosed with COVID-19 infection. Nurses should be supported physically, socially and psychologically. They should be appropriately rewarded, so that they can willingly provide care to patients if a similar situation occurs in the future. Support programmes should be established for nurses to cope with the psychosocial problems they experience when caring for patients diagnosed with COVID-19.

#### 4.4.3. Context: In Which Context Will the Framework Be Implemented?

Context is the third concept on the survey list of Dickoff et al. (1968) [5]. Context is the setting in which the activity of psychosocial support for nurses caring for patients diagnosed with COVID-19 infection will take place. The conceptual framework for the psychosocial support of nurses caring for patients diagnosed with COVID-19 infection has been developed within a context by an agent and received by the recipient. The context of the framework includes the four public hospitals caring for patients diagnosed with COVID-19 infection in NWP. The contexts of this study are vital components, because this is where the nurses who are caring for COVID-19 patients can be found.

#### 4.4.4. Procedure: How Will the Framework Be Implemented?

Procedure is the fourth concept on the survey list of Dickoff et al. (1968) [5]. Procedure refers to the steps or processes followed to ensure the success of the development of the psychosocial support guidelines for the nurses caring for patients diagnosed with COVID-19 infection. The procedure followed the following two steps: mobilising effective managerial support and providing adequate healthcare resources.

Step 1: Mobilising effective managerial support for nurses caring for patients diagnosed with COVID-19 infection

Mobilising effective managerial support for nurses is significant, because COVID-19 has had negative impacts on nurses, fostering fear, depression and anxiety. This indicates the necessity of providing support to nurses caring for COVID-19 patients. All the agents involved should support nurses. The support to nurses should include continuous in-service training for all nurses caring for COVID-19 patients. Therefore, nurses should be informed about COVID-19 and its management practices, including the usage of equipment such as ventilators. 

Management should ensure that psychological support for nurses is readily available, and where necessary, nurses should be referred to occupational health and safety (OHS) for investigations. Nurses should be provided with immune boosters to boost their immune systems. Hospital management should visit the ward regularly to support nurses and show appreciation for the work nurses are performing. 

Step 2: Provision of adequate healthcare resources to nurses caring for patients diagnosed with COVID-19

The provision of adequate human and material healthcare resources is imperative in the process of providing psychosocial support for the nurses caring for patients diagnosed with COVID-19 infection. A major issues facing nurses is a critical shortage of nurses, beds and medical supplies, including personal protective equipment (Al Thobaity and Alshammari, 2020) [17]. Providing human resources with an adequate number of nurses will reduce the workload of nurses. Governments and health policymakers need to invest in nurses and equipment and to pay attention to the needs of the health systems to ensure a healthy population (Turale and Nantsupawat, 2021) [29]. 

Nurses have a fear of being infected with COVID-19; therefore, they need to be provided with proper PPE such as protective gowns, masks and gloves to render quality patient care. All permanent posts should be filled, so that nurses who are on contracts can apply to obtain occupational and financial security.

#### 4.4.5. Dynamics: What Are the Dynamics of the Framework?

The fifth concept on the survey list of Dickoff et al. (1968) [5] is dynamics. Dynamics are the motivating factors or activities that will occur to ensure the success of the developed conceptual framework for the psychosocial support of nurses caring for patients diagnosed with COVID-19 infection. The motivating factors for the conceptual framework developed include in-service training, occupational and financial security, teamwork, motivational programmes, the provision of immune boosters and follow-up treatment for nurses. The agents are expected to engage nurses to facilitate psychosocial support. There is a need for motivation and the provision of a safe environment for nurses. The collaborative efforts of the agents are important in sustaining psychosocial support for nurses caring for COVID-19 patients.

#### 4.4.6. Terminus: What Will Be the Endpoint of the Framework?

This is the sixth and last concept on the survey list of Dickoff et al. (1968) [5]. Terminus, in this context, refers to the endpoint or the outcome of the newly developed conceptual framework. The endpoint or the aim of the newly developed framework discussed here is to support the nurses caring for patients diagnosed with COVID-19 infection in NWP. The conceptual framework will enable nurses to develop effective coping skills. The implementation of the developed conceptual framework should be supported by all agents. 

## 5. Limitations

The study was conducted in four public hospitals in the NWP, South Africa. Furthermore, the development of the conceptual framework was based on a qualitative study. Thus, the conceptual framework cannot be generalised; rather, it can be transferred or applied in other provinces or different contexts. 

## 6. Conclusions

A qualitative, descriptive phenomenological, contextual research design was used to develop a conceptual framework for the psychosocial support of the nurses caring for patients diagnosed with COVID-19 infection in NWP, South Africa. The conceptual framework consists of important information regarding the psychosocial support of nurses, such as the agents, recipients, context, procedures, dynamics and terminus involved. The developed framework can provide solutions for healthcare institutions to provide psychosocial support to nurses caring for patients diagnosed with infectious diseases such as COVID-19.

## Figures and Tables

**Figure 1 ijerph-20-05078-f001:**
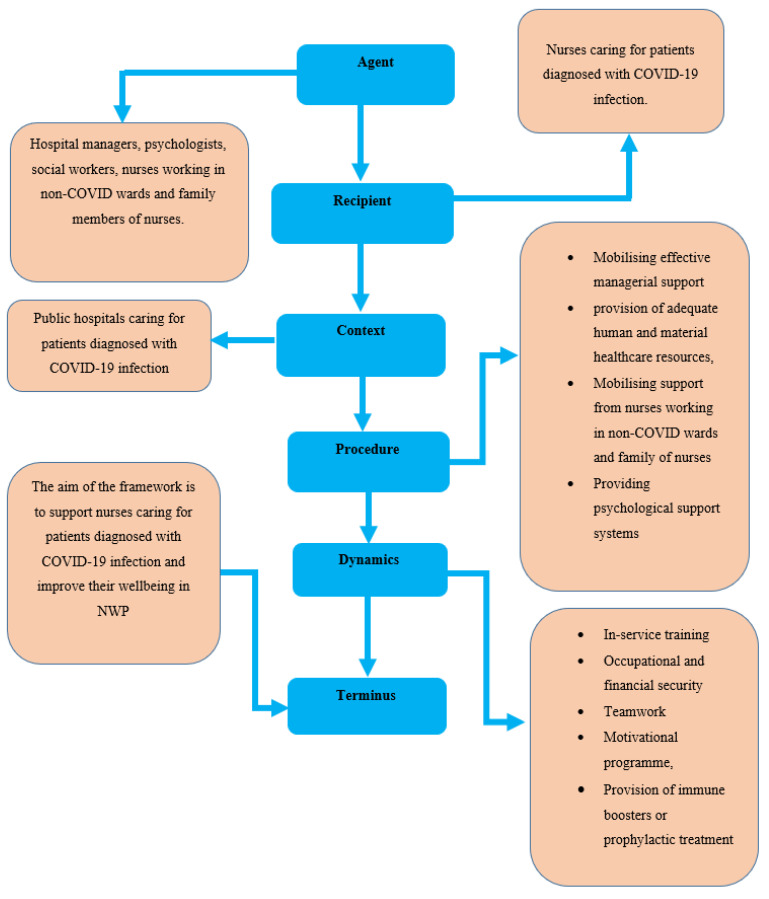
A conceptual framework for the psychosocial support of nurses caring for patients diagnosed with COVID-19 infection in NWP adapted from Dickoff et al. (1968) [5].

**Table 1 ijerph-20-05078-t001:** Information on studies included in integrative literature review (Dikobe et al., 2023) [1].

Author(s) Year Country	Design and Method Population and Sample	Purpose	Quality Appraisal (Scale: High = h, Low = l nr = Not Reported
Alnazly et al., 2021 [16], USA	Cross-sectional, correlational design to collect data N = 365	To assess the respective levels of fear, anxiety, depression, stress and social support and the associated factors experienced by Jordanian healthcare workers during the COVID-19 pandemic.	(h) Aims and objectives clearly stated (h) Study design adequately described (h) Research methods appropriate (nr) Explicit theoretical framework (h)Limitations presented (h) Implications discussed
Al Thobaity and Alshammari, 2020 [17], Saudi Arabia	Integrative review N = 95	To explore the issues facing nurses during their response to the COVID-19 crisis.	(h) Aims and objectives clearly stated (h) Study design adequately described (h) Research methods appropriate (nr) Explicit theoretical framework (h) Limitations presented (h) Implications discussed
Chew et al., 2020 [18], Singapore	Narrative synthesis N = 144	To synthesise extant literature regarding the combined psychological responses and coping methods used by the general population in past outbreaks.	(h) Aims and objectives clearly stated (h) Study design adequately described (h) Research methods appropriate (nr) Explicit theoretical framework (h) Limitations presented (h) Implications discussed
Fang et al., 2021 [19], China	Cross-sectional study design N = 540	To understand the mental health status and needs of healthcare workers, so as to provide a scientific basis for alleviating their psychological pressure.	(h) Aims and objectives clearly stated (h) Study design adequately described (h) Research methods appropriate (nr) Explicit theoretical framework (h) Limitations presented (h) Implications discussed
Muller et al., 2020 [20], Norway	Systematic review N = 59	To identify, assess and summarise research on the mental health impact of the COVID-19 pandemic on HCWs.	(h) Aims and objectives clearly stated (h) Study design adequately described (h) Research methods appropriate (nr) Explicit theoretical framework (h) Limitations presented (h) Implications discussed
Rathnayake et al., 2021 [21], Sri Lanka	Colaizzi’s phenomenological approach N = 14	To explore the experiences and challenges of nurses who have worked with patients hospitalised with COVID-19.	(h) Aims and objectives clearly stated (h) Study design adequately described (h) Research methods appropriate (nr) Explicit theoretical framework (h) Limitations presented (h) Implications discussed
Sehularo et al., 2021 [22] SA	Narrative literature review	To explore and describe the coping strategies used by nurses during the COVID-19 pandemic.	(h) Aims and objectives clearly stated (h) Study design adequately described (h) Research methods appropriate (nr) Explicit theoretical framework (h) Limitations presented (h) Implications discussed
Shatri et al., 2021 [23] Indonesia	Evidence-based clinical review N = 6	To identify psychotherapy as a psychological health intervention for healthcare workers during the COVID-19 pandemic.	(h) Aims and objectives clearly stated (h) Study design adequately described (h) Research methods appropriate (nr) Explicit theoretical framework (h) Limitations presented (h) Implications discussed.
Xu et al., 2021 [15] China	Interventional clinical observation and psychological evaluation N = 1198	To evaluate the mental health of hospital staff before and after a psychological intervention by the Employee Assistance Program (EAP).	(h) Aims and objectives clearly stated (h) Study design adequately described (h) Research methods appropriate (nr) Explicit theoretical framework (h)Limitations presented (h) Implications discussed

**Table 2 ijerph-20-05078-t002:** Biographical data of participants for the FGDs. Dikobe et al. (2023) [24].

VARIABLE	CATEGORIES	NUMBER
Gender	Male	2
	Female	24
Age in years	33–60	
Nurses participating in the study per institution	Hospital A	8
	Hospital B	5
	Hospital C	7
	Hospital D	6
Current position of nurses Occupational categories	Professional nurses (PN)	10
	Enrolled nurses (EN)	7
	Enrolled nursing auxiliary (ENA)	9
Educational level	Certificate	16
	Diploma	6
	Degree	4
Years of experience in the above work position	All categories of nurses	3–29

## Data Availability

Data sharing does not apply, as no new data were discovered and/or analysed.

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
