# Peer review of "Conceptual Framework for the Psychosocial Support of Nurses Caring for Patients Diagnosed with COVID-19 Infection in North West Province, South Africa"

_ijerph, 2023, doi:10.3390/ijerph20065078_

Round 1
Reviewer 1 Report
A very innovative and insightful approach to some of the challenges posed by COVID-19
However, prior to publication the authors should address the following:
1) Grammar and typo errors
2) The 14 references utilised in this paper are not adequate, considering that an integrative review was conducted as the basis of this paper.
3) More specific sources should be cited in the integrative literature review instead of general references.
4) For the integrative review it is recommended that the authors giving an outline of the literature reviewed: showing the author & year of publication, study setting, sample, research design and findings.
5) It is also unclear which aspects of this study are essential in informing critical aspects of this paper. The findings are too summarised, and authors seem to assume that the reader has an in-depth understanding of the background to the study that informed the development of this paper.
Reviewer 2 Report
This is an important, timely topic.
This paper is reporting the results of a literature review, a qualitative study, and the development of a conceptual framework. It seems that the literature review led to the development of the conceptual framework. There are a lack of details with respect to the literature review. The themes are presented, but there are no description of the studies to support the themes. In addition, a qualitative study with focus groups was done with nurses, however, the results of these focus groups are not described.
The introduction to the paper is focused on the definitions of a conceptual framework. It would make sense to include a focus on psychosocial support for nurses, and the impact of the COVID pandemic on nurses.
Authors’ names are missing in the introduction. Instead of presenting the reference number, suggest using the authors’ names that are being referred to.
In the methods section, it is not clear how nurses were recruited to participate in focus groups. The details about the participants, such as ages, would fit better in the results section.
The methods sections could include details about the questions used with the focus groups.
Under trustworthiness (section 2.2.2) the authors refer to transferability. Transferability typically refers to being able to apply the findings to other settings or groups.
The description of the literature review is too short, and is missing references.
The results section is focused on the conceptual framework, however it is not clear how the literature review and the qualitative study led to the development of the concepts that are described.
Instead of combining the results & discussion sections, they could be provided separately. This would make the differences between the results and the discussion more clear.
It might be worthwhile to consider using the integrative literature review and the conceptual model as the background.
Suggest referring to South Africa in the title, as most readers will not know which North West Province the authors are referring to (as North West is used for other provinces or territories in other countries.)
Reviewer 3 Report
The theoretical framework is very weak.
The article is not edited, e.g.
- Page 1, line 60: The author's name should be given, followed by the reference number. Likewise in many other places.
- The drawing is for the signature and the whole thing is illegible.
The literature contains very few bibliographic items.
Some bibliographic items are not of a scientific nature (e.g. item 6).
Practically no conclusion in the article.
Round 2
Reviewer 1 Report
The authors have attended to the comments/concerns raised by the reviewer and the manuscript is much improved
Author Response
.
Reviewer 2 Report
The authors have made a number of changes and this is a stronger paper.
The second sentence refers to COVID-1. Do the authors mean COVID-19?
The study design (page 2, line 90) refers to “empirical”. Empirical what? If the authors mean the qualitative study, this should be stated.
The authors could consider publishing this as 2 separate studies. The literature review could be its own paper. If the authors wish to keep the material together, it makes more sense to include the literature review in the background section.
If the literature review is to stay part of this study, it needs much more description in the methods as to how it was carried out. Again, as it currently stands, it would fit better in the background section.
It’s not clear how 2 phases of the qualitative study took place. There is a description of the focus groups which would be one phase. If there was another step of data collection, this could be included. There are only brief details given of the findings from the focus groups. The results section seems to focus on the findings of the literature review. Again, this is confusing, and I would suggest considering 2 separate papers rather than trying to merge all of the findings together.
On page 8, line 232 refers to the development phase, and it is not clear what this is. It seems to be the development of the conceptual framework.
The discussion section is interesting and has many important points. However, the results section is comparatively short, and it is not clear how the results section led to the comprehensive discussion section.
Reviewer 3 Report
There have been many positive changes in the current version of the article. I appreciate the expanded literature review. Chapter 2 Material and Methods and also Chapter 3 Results have been significantly expanded. Proofreading and editing of the article has also been done. All this has dispelled my earlier serious doubts about the paper.
Author Response
.